# Relationships between Socioeconomic Indicators and Motor Performance of Schoolchildren in Slovakia

**DOI:** 10.3390/ijerph182413153

**Published:** 2021-12-13

**Authors:** Tomas Peric, Pavel Ruzbarsky, James J. Tufano

**Affiliations:** 1Faculty of Sports, University of Presov, 08001 Presov, Slovakia; pavel.ruzbarsky@unipo.sk; 2Faculty of Physical Education and Sports, Charles University, 16252 Prague, Czech Republic; tufano@ftvs.cuni.cz

**Keywords:** physical activity, motor performance test, first grade, socioeconomic status, poverty

## Abstract

Physical inactivity is currently a significant problem in Western societies. Among the many factors that affect the amount of physical activity in children, socioeconomic status, the type of school, and the area where they live can play a major role in physical development. A total of 31,620 children (6.3 ± 0.53 years; 15,726 boys; 15,893 girls), representing 55.6 % of the entire Slovak population of first graders, participated in eight motor performance (MP) tests, the results of which were correlated with somatic parameters (body height, body weight, and body mass index (BMI)) and socioeconomic indicators (SEIs) within the children’s respective territorial regions. The largest correlations were observed between MP and unemployment rate (r = −0.802) and between MP and economically active population with primary education (r = −0.807). Other correlations between MP and SEIs were also found at the level of self-governing regions (r = 0.995) and districts (r = 0.551). SEIs and MP were strongly correlated, indicating that children who grow up in better socioeconomic environments display greater MP. Therefore, national and local governments should provide tangible and intangible resources to enable the proper development of children’s MP.

## 1. Introduction

In 2015, the Slovak government passed a new Sports Act that requires nationwide assessment of children’s physical abilities in grades 1 and 3 in elementary schools. This testing took place in the Slovak Republic for the first time in 2018 and should subsequently be implemented every year, which provides a unique opportunity to evaluate various aspects of motor performance in children on a large scale. Thus, research on such a large dataset allows for an extensive analysis, which becomes the basis for analyses in other studies, not only within the Slovak Republic, but also internationally. In this analysis, we attempted to identify the relationships between the level of physical performance of primary school pupils in the Slovak Republic and the socioeconomic indicators of individual regions or districts, and to point out similar findings abroad.

Physical inactivity is currently a significant problem in Western societies and has a significantly negative impact on children. It can manifest in various ways, such as elevated body weight, which can lead to childhood obesity, which can further lead to non-communicable diseases during adulthood, such as cardiovascular diseases (hypertension, ischemic heart disease), diabetes, and others [1,2,3]. Besides these health problems, overweight children can also experience psychological and social problems, as obese children tend to congregate in social groups with others who are less physically skilled, which can lead to feelings of inferiority, seclusion, and even bullying [4,5]. Over time, these events can have a snowball effect, resulting in a constant cycle of less activity, worse motor performance, and poorer health. Therefore, it is necessary to intervene with measures to increase physical activity in the early stages in life.

Among the many significant factors that affect the amount of physical activity children engage in, socioeconomic status, the type of school, and the area where the children grow up are likely to play major roles. In support of this, one study indicated that socioeconomic status (SES), the amount of physical activity, and motor performance (MP) were all highly correlated in German children [6]. Specifically, in that study, younger children, children of immigrants, and children with lower SES were less physically active and had worse MP compared to children of non-immigrants and children with greater SES [6]. Such patterns have also been observed when comparing rural and urban children [7,8], and research also suggests that the differences in MP among children of the same nationality are caused primarily by socioeconomic inequalities between a country’s regions [9]. Although these findings are important for identifying potential areas of improvement for national and local governments, the sample sizes used in many previous studies only represent a small percentage of the overall population (e.g., 3664 males and 4130 females answered questions on physical activity and social status at 14 and 31 years of age [10], which at first may seem like a large sample, but it is not a large percentage of the overall available population). Furthermore, a country’s culture or natural environment may encourage or inhibit physical activity, indicating that more large-scale studies are required to shed light on the effects of various SES on MP in different regions of the world. 

In the context of the theoretical background and the literature sources cited therein, it would be very interesting to compare the results of these studies (conducted mainly in Western Europe) with the results of a baseline population for a given population and year in a relatively “new” country of the European Union (which generally includes countries with a lesser socioeconomic status than Western European countries).

Specifically, the aim of this study was to clarify the relationships between various SEIs in different regions of Slovakia and their importance for the MP of children who are not yet significantly affected by school and out-of-school physical activity (i.e., young children who may benefit from any data-driven recommendations that can change society). 

## 2. Materials and Methods

### 2.1. Participants

Because of significant legislative changes in the Slovak Republic, a new law on sports and physical activity was adopted, which included mandatory testing of motor preconditions for all of the country’s first graders [11]. As a result, a total of 31,620 children (6.3 ± 0.53 years old; 15,726 boys; 15,893 girls; 1 gender unspecified) were divided into groups, according to the territorial division of the country, reflecting 8 self-governing regions and 79 districts. The number of districts in each self-governing region ranged from 7 to 13, which is based on population and land area. However, the land area and population size of the self-governing regions and the districts were not taken into account in this study, as they are largely political in nature. All participating children were included in the correlation analyses between SEIs and MP. 

### 2.2. Motor Performance Tests

All children were assessed using the following 8 motor performance tests that evaluate various physical abilities and skills: coordination—repeated stick routine (sec); manipulative coordination—rolling 3 balls (sec); agility—4 × 10 m shuttle run (sec); explosive power—standing broad jump (cm); upper body muscular strength—bent-arm hang test (sec); endurance in strength—sit-ups in 1 min (rep); endurance—20-metre shuttle run (repetitions); and flexibility—sit-and-reach test (cm). All testing was conducted for each school during a single morning by their physical education teachers. The sequence of the tests and standardised warm-up was defined by the testing protocol. The actual testing was preceded by a standardised warm-up period of 10 min, and the tests were ordered into three blocks, as follows, for the evaluation: (1) body height and body weight, sit-and-reach test, repeated stick routine, rolling 3 balls, standing broad jump, bent-arm hang test, sit-ups in 1 min; (2) 4 × 10 m shuttle run; (3) 20-metre shuttle run [12]. Special software was developed by the Ministry of Education of the Slovak Republic for testing purposes with defined control mechanisms that were used to eliminate recording errors or incorrectly recorded data by teachers [13]. To create an overall MP score, the crude scores were normalised using a T-score $(T=50+10\cdot (\frac{\left(x_{i}-\bar{x}\right)}{s}$)) (explanation: *x_i_* the child’s individual test result; *x*—the arithmetic average of the group in a given test; s—standard deviation of the group in a given test). The results of each test (T-score) were then summed up to create the overall MP T-score. The reliability of the motor performance tests ranged from 0.80 to 0.97 [14,15]. 

### 2.3. Regional Data

Data for the self-governing regions and districts were obtained from the Statistical Office of the Slovak Republic [16]. The SEIs that were available for self-governing regions included the following: unemployment rate (%); incomes and living conditions of the households, such as the average gross equivalent household income (EUR) and the average disposable equivalent household income (EUR); the economically active population based on education, such as those with a university education (%) and with primary education (%); and regional gross domestic product (EUR). The SEIs assessed at the level of districts included the following: unemployment rate (%); average nominal monthly earnings (EUR) based on gender; and the number of companies in the district.

### 2.4. Data Analyses 

Data were analysed using descriptive statistics (mean, standard deviation, minimal and maximal values). The results of the motor performance tests were analysed using correlation analysis in two steps. First, each individual SEI was correlated with the overall MP variable as a sum of T-scores from each of the eight individual motor performance test results; second, multiple correlations of all SEIs were carried out with the overall MP variable.

The data were analysed for the whole sample and for each gender (one child without gender specification was included in the whole sample analyses). Analyses were conducted using SPSS version 24.0 (IBM, New York, NY, USA). The data were also analysed at two territorial unit levels: at the level of self-governing regions and at the level of districts within those self-governing regions. 

## 3. Results

The descriptive characteristics of the sample separated by gender are provided in Table 1 [17]. The results presented are for information only and were used as a basis for further analysis. It was not the purpose of these descriptive data to assess the quality of fitness of children at a given age.

The main aim of analysis was the assessment of SES as an independent variable and overall MP (T-scores) as a dependent variable. The data published by the Statistical Office of the Slovak Republic are different for different territorial units, i.e., there are six indicators available for the level of self-governing regions and only three for the level of districts. The question arose whether it was appropriate to compare SEIs at the district level because they could duplicate the level of SEIs in the self-governing regions. However, data obtained from the Statistical Office of the Slovak Republic showed that, even within one self-governing region, differences in SEIs between districts could be significant. The results of the correlation analyses between individual SEIs and overall MP (T-scores), at the level of self-governing regions, are provided in Table 2.

The results of the correlation analyses between individual SEIs and overall MP (T-scores) at the level of districts are provided in Table 3. 

## 4. Discussion

Based on these results, it is possible to conclude that, at the district level, there is relatively high agreement between SEIs (clustered in one variable, which emerged as a latent variable in the multiple correlation calculation procedure) and the overall MP (T-scores) of the children. It is necessary to note that only those SEIs mentioned were freely available at the district level. It is highly probable that if there were other indicators available, the correlations could have been even higher. Based on this, it may be concluded that the economic status of the family can greatly impact the child’s ability to participate in formal sport activities. If the family does not have enough financial resources, it may be reflected in the overall physical level of the children in that district such that they may have a lower level of MP.

At the level of self-governing regions, there is absolute agreement between SEIs (clustered into a single variable) and the overall MP of the children. Therefore, it may be concluded that, in agreement with other studies [18,19], the economic status of the family plays a large role in all aspects of MP of the children (e.g., organised sport—in clubs; unorganised sport—with parents, friends, on your own). Thus, if the family does not have enough financial resources, it can be reflected in the overall physical level of the children in that region by a lower level of MP. Furthermore, it is possible to conclude that overall MP is significantly correlated with *unemployment rate* and *economically active population with primary education*. Those SEIs show a high influence on overall MP (compare [20]). Similar results, but in a different context (focused on lifestyle and on the socioeconomic status of the family), were presented in another study [21].

Although the data presented within this article are valuable, as they confirm that SEIs are related to the MP of children in the area, other factors should also be considered, such as the influence of immediate and extended family members. In support of this, previous research has shown that the family environment fundamentally influences many areas that can also influence the MP of children [22,23,24]. Among these family environmental factors, the following should be assessed in future research, as they likely influence physical activity, sport participation, and MP: the daily regime of the children, the value of sports and physical activities to the family, the number of children in the family, the financial resources of the family (economic status), logistical resources of the family (number of adult members in the broader family), and the conditions for sport (sport infrastructure) in the neighbourhood.

The multiple correlation analysis in the present study clearly identified a relationship between MP and SEIs, specifically between a higher MP of children and a higher socioeconomic status in the region. 

If physical activity and sport are among the preferred values in the family [25,26,27] (and are supported by the other above-mentioned areas as well), then the children are unlikely to experience the negative consequences of physical inactivity (low level of MP, increased body weight, etc.). However, the crucial question arises in a situation where the family does not understand the importance of sport and physical activity for the development of the children, and when the family perceives physical activity as being insignificant for their development. Most often, these beliefs are held by those of a lower socioeconomic status, where the available financial resources do not allow for (or allow for, but to a very limited extent) children to be involved in commercially run physical activities. In these instances, involvement of the children can be limited by a) participation costs, e.g., club fees, costs of equipment, and training camps, and b) transportation costs, particularly to the more distant sport facilities when it becomes necessary to use their own car. In this case, the transportation costs could be higher than participation costs, and it would not be possible to secure the children’s participation organisationally and logistically. This situation becomes even more important when the child lives with only one parent and cannot access help from broader family members (e.g., grandparents or other members), and when there are other young dependent children who require adult supervision. The most important factor is the situation where the parent does not have the opportunity to adapt his or her work schedule to the needs of the children.

It is apparent that the above-mentioned negative situations may appear in various combinations. However, there is yet another aspect which could be mentioned while addressing the above situations, that is, the long-term influence of physical inactivity on each individual family, village, or locality. There is some assumption that children with a lower level of MP have transferred this “expression” as a certain “epigenetic” determination from their parents [28,29,30]. It is very likely that children living in families in which sport and movement have little importance within the family values will transfer this “(in)value habit” to their children. In this way, the trend will be deepened with each generation to come. 

Therefore, a fundamental challenge becomes how to secure sufficient physical activity for children, when the family does not want to or cannot do so. The solution lies outside the scope of this article, but it is now a problem that needs to be addressed at the levels of villages, cities, districts, self-governing regions, and government [31,32,33]. Our findings corroborate those of several other studies which suggest that socioeconomic status plays an important role in the overall health of the population [34].

The main limitation of this study was the method of data collection. As the data were based on a group of all pupils in their first year in primary schools in the Slovak Republic, the testing had to be carried out by the physical education teachers themselves. This limitation was minimised by selecting motor tests that have high reliability and are simple for children to understand, and by having software support to avoid errors in recording the measured data. In selecting the motor tests, we were limited by the spatial arrangements in the school gymnasium, as we wanted to ensure comparable conditions, and by the total testing time. Another limitation was the time constraint of data acquisition, which was limited by a uniform testing schedule of 4 weeks. Another limitation of the study was the scope and content of the socioeconomic indicators, which were based on publicly available data provided by the Statistical Office of the Slovak Republic.

## 5. Conclusions

The large dataset of the present study allowed for extensive analysis, which can serve as the base for other analyses not only in the Slovak Republic but also internationally.

The data from these tests were subjected to correlation analyses with SEIs. A high correlation between those indicators and MP (T-scores) was observed. This may lead to various questions related to health and politics. One of the main questions is whether the state should provide development of children’s MP, which is, in principle, family bound.

Based on that, it is possible to formulate the following conclusions:-Children do not engage in sport spontaneously; physical activity by children is limited (involving play with other children outside), and it is minimal in the daily regime of children;-If children engage in leisure-time physical activity, it is under the supervision of adults as a family activity (e.g., trips, visits to playgrounds, sport activities performed together, organised sport activity in after-school sport activities, or in sport clubs).

## Figures and Tables

**Table 1 ijerph-18-13153-t001:** Descriptive characteristics of the sample—boys and girls separately.

Variables	Boys (*n* = 15,726)				Girls (*n* = 15,892)			
	AVG	STD	Min.	Max.	AVG	STD	Min.	Max.
Age (years)	6.3	0.53	5.0	11.0	6.4	0.57	4.0	10.0
Body height (cm)	122.5	6.02	89.0	155.0	123.9	6.13	95.0	157.0
Body weight (kg)	24.2	4.94	13.0	70.0	24.9	5.08	14.0	84.0
BMI (kg·m^−2^)	16.0	2.48	10,1	43.4	16.2	2.46	10,1	42.3
Sit-and-reach test (cm), /r_stab_ = 0.97/	6.1	6.88	−50.0	50.0	3.9	6.96	−50.0	50.0
Rolling 3 balls (sec), /r_stab_ = 0.84/	37.8	14.89	10.0	198.0	34.2	13.89	10.0	170.0
Standing broad jump (cm), /r_stab_ = 0.93/	102.2	18.79	25.0	228.0	110.7	19.96	11.0	232.0
Bent-arm hang test (sec), /r_stab_ = 0.80/	5.7	6.80	0.0	99.0	6.8	7.62	0.0	99.0
Repeated stick routine (sec), /r_stab_ = 0.95/	36.6	12.92	10.0	99.0	36.8	13.35	10.0	99.0
Sit-ups in 1 min (rep), /r_stab_ = 0.80/	20.6	8.48	0.0	60.0	21.9	8.72	0.0	77.0
4 × 10 m shuttle run (sec), /r_stab_ = 0.90/	16.4	3.50	10.1	73.8	15.8	3.31	10.1	77.0
20-metre shuttle run (*n*), /r_stab_ = 0.93/	14.3	6.84	1.0	40.0	16.0	7.89	1.0	40.0
Overall motor performance (T-score)	394.7	40.82	116.8	603.7	405.4	42.89	123.6	621.9

Table legend: AVG—the arithmetic average; STD—standard deviation; Min—minimum value in the group; Max—maximal value in the group; r_stab_—the reliability (stability) coefficient of the test.

**Table 2 ijerph-18-13153-t002:** Results of correlation analyses between SEIs and overall MP (T-scores), and of multiple correlations, at the level of self-governing regions.

Variable	All	Boys	Girls
**Unemployment rate (%)**	−0.802 **	−0.717 *	−0.847 **
**Average gross equivalent household income (EUR)**	0.495	0.450	0.524
**Average dispo** **sable equivalent household income (EUR)**	0.529	0.485	0.557
**Economically active population with university education (%)**	0.194	0.180	0.207
**Economically active population with primary education** **(%)**	−0.807 **	−0.749 *	−0.819 **
**Regional gross domestic product (EUR)**	0.408	0.373	0.431
**Multiple correlation**	0.995 **	0.988 **	0.999 **

* *p* = 0.01. ** *p* = 0.05.

**Table 3 ijerph-18-13153-t003:** Results of correlation analyses between SEIs and overall MP (T-scores), and of multiple correlations, at the level of districts.

Variable	All	Boys	Girls
**Unemployment rate (%)**	−0.541 **	−0.433 **	−0.588 **
**Average nominal monthly earnings (EUR)**	0.378 **	0.426 **	0.340 **
**Number of companies in the district**	0.234 *	0.300 **	0.215 *
**Multiple correlation**	0.551 **	0.488 **	0.595 **

* *p* = 0.01. ** *p* = 0.05.

## Data Availability

The data are uploaded to a public server for public use (can be found here: https://www.testovanieziakov.sk/content/files/NSC_NP_vysledky.pdf, accessed on 18 June 2021).

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
