# Peer review of "Relationships between Socioeconomic Indicators and Motor Performance of Schoolchildren in Slovakia"

_ijerph, 2021, doi:10.3390/ijerph182413153_

Round 1

Reviewer 1 Report

Dear authors,

In spite of your study including large sample size and it is very relevant and novelty for your country. However, this study is lacking in quality and consistency, mainly, the methodology, results, and discussion sections.

Please try to change the words "traditional" in line 64 and "new" in line 65, using scientific language more properly.

The material and methods section is confusing and not well explained. For instance, in the participants' section, I would recommend you describe better your sample. Please add in line 75 a description in the same way that you describe in the abstract section “(6.3 ± 0.53 years; 15,726 boys 15,893 girls)”. The 2.2. Motor performance tests need a reference with English languages explanation to improve a better comprehension of readers. Besides, the overall MP score would need well-explained. I have some doubts about how T-score was calculated. You have to explain the letters of the formula shown in line 90. Please, use the overall MP T-score in your manuscript not only MP or overall MP. These terms should be unified in all text.

Furthermore, I suggest adding the footnotes, describing abbreviations used in table 1.

Additionally, I suggest improving the redaction of your discusión section. Please delete “Based on these results, it is possible to conclude that” in the first paragraph, and in a similar way, at the beginning of the next one, and try to add some references to justify your statements.

I suggest changing the word “impact” in line 165, for “is related to..”Because there is no cause-effect by your statistical analysis. And please, try to reformulate the paragraph from line 177 to 184 or add it as conclusions in the last section of your manuscript.

Finally, you conclude your discusión section by saying that “The solution reaches beyond the frame of this article, but at the moment, it is a problem that needs to be addressed at the levels of the villages, cities, districts, self-governing regions, and government” However, I would like to suggest you mention the different levels that we should manage the exercise adherence and physical activity promotion at different settings. For instance, according to the WHO's five dimensions adherence model (i.e.: Calonge Pascual S, Casajús Mallén JA, González-Gross M. Adherence Factors Related to Exercise Prescriptions in Healthcare Settings: A Review of the Scientific Literature. Res Q Exerc Sport. 2020 Sep 9:1-10. doi: 10.1080/02701367.2020.1788699), or by a social ecological approach (i.e. Burke, S., Utley, A., Belchamber, C., & McDowall, L. (2020). Physical Activity in Hospice Care: A Social Ecological Perspective to Inform Policy and Practice. Res Q Exerc Sport, 1-14. doi: 10.1080/02701367.2019.1687808), adding scientific value to your final statement in your discussion section.

Yours faithfully,

Author Response

Please try to change the words "traditional" in line 64 and "new" in line 65, using scientific language more properly. 

The text has been modified according to the opponent's requirements

Please add in line 75 a description in the same way that you describe in the abstract section “(6.3 ± 0.53 years; 15,726 boys 15,893 girls)”. 

The text has been modified according to the opponent's requirements

The 2.2. Motor performance tests need a reference with English languages explanation to improve a better comprehension of readers. 

The text has been modified according to the opponent's requirements

Besides, the overall MP score would need well-explained. I have some doubts about how T-score was calculated. You have to explain the letters of the formula shown in line 90. Please, use the overall MP T-score in your manuscript not only MP or overall MP. These terms should be unified in all text.

The text was modified according to the requirements of the opponent in the text, where this change was justified. 

Furthermore, I suggest adding the footnotes, describing abbreviations used in table 1.

The text has been modified according to the requirements of the opponent.

Please delete “Based on these results, it is possible to conclude that” in the first paragraph, and in a similar way, at the beginning of the next one, and try to add some references to justify your statements.

The text has been modified according to the requirements of the opponent.

I suggest changing the word “impact” in line 165, for “is related to..”Because there is no cause-effect by your statistical analysis. And please, try to reformulate the paragraph from line 177 to 184 or add it as conclusions in the last section of your manuscript.

The text has been modified according to the requirements of the opponent.

Finally, you conclude your discusión section by saying that “The solution reaches beyond the frame of this article, but at the moment, it is a problem that needs to be addressed at the levels of the villages, cities, districts, self-governing regions, and government” 

The text has been modified according to the requirements of the opponent.

Reviewer 2 Report

1. On line 11 you need a punctuation mark between boys and girls.
2. At the end of the introduction, the purpose of the study is not indicated.
3. Why does the title of section 2 mention the material?
4. Why separate the results sections from the discussions? I think. it is necessary to discuss as the results are presented.
5. Why is there only one output, only on 6 lines? This greatly impoverishes the achievement of the manuscript.

Author Response

On line 11 you need a punctuation mark between boys and girls.

The text has been modified according to the requirements of the opponent.

At the end of the introduction, the purpose of the study is not indicated.

The text has been modified according to the requirements of the opponent.

Why does the title of section 2 mention the material?

The title of the chapter has been modified according to the requirements of the journal.

Why separate the results sections from the discussions? I think. it is necessary to discuss as the results are presented.

The chapters are separated according to the requirements of the journal.

Why is there only one output, only on 6 lines? This greatly impoverishes the achievement of the manuscript.

The text has been modified according to the requirements of the opponent.

Reviewer 3 Report

Dear authors,
Thank you for the opportunity to evaluate this study. Undoubtedly, it raises a very important health problem, which is supporting the development of the sport of children. As the authors themselves observed, a healthy lifestyle is a guarantee of health. Therefore, it is extremely important to cultivate the principles of a healthy lifestyle in children.
My suggestions for the article include:
1) Lack of information on authors' contributions, ethics committee approval and data availability.
2) Lack of information about the project, i.e. it was a one-time program, the need to test every first grader?
3) In the introduction, there are no research hypotheses and no clear purpose of the work, which have been supported by the literature.
4) Is the law modelled on other countries? Somewhere similar / the same research on children is being conducted?
5) There are no work limits in the discussion, of which there are undoubtedly a lot - please think over and fill in a separate paragraph.
6) What do the above results mean? Condition is good, bad? References to other works? Comparison with other countries.

Author Response

Lack of information on authors' contributions, ethics committee approval and data availability.

These parts and information were completed according to the requirements of the opponent.

Lack of information about the project, i.e. it was a one-time program, the need to test every first grader?

The text has been modified according to the requirements of the opponent.

In the introduction, there are no research hypotheses and no clear purpose of the work, which have been supported by the literature.

The aim of the work was added at the end of the Introduction section.

Is the law modelled on other countries? Somewhere similar / the same research on children is being conducted?

This testing system was created as specific to the needs of the Slovak Republic and has not been modified according to other countries. 

There are no work limits in the discussion, of which there are undoubtedly a lot - please think over and fill in a separate paragraph.

The text has been modified according to the requirements of the opponent.

What do the above results mean? Condition is good, bad? References to other works? Comparison with other countries.

The text has been modified according to the requirements of the opponent.

Reviewer 4 Report

This research has an original objective and a meaningful content. I congratulate the authors for the work done. I am grateful with the editors for the possibility of revising this manuscript. Although the quality of the manuscript is high, I would like to make some contributions that I hope will increase it and improve readers' understanding. 

Introduction:

The introduction is clear and well worked.

Material and methods:

  • Lines 73-75: Authors must include citation about new law on sports and physical activity adopted  in Slovak Republic.
  • Lines 86-87: Authors explain that "The sequence of the tests and standardised warm-up was defined within the standardisation of the testing protocol [11]", but reference is in Slovak. I suggest including a little explanation about protocol.
  • Line 87: What software?
  • Table 1: Table legend must be include to better understanding. 

Data analysis:

The statistical analysis is correct and well described.

Discussion:

It is well oriented.

Author Response

  • Lines 73-75: Authors must include citation about new law on sports and physical activity adopted  in Slovak Republic.
    • The text has been modified according to the requirements of the opponent.
  • Lines 86-87: Authors explain that "The sequence of the tests and standardised warm-up was defined within the standardisation of the testing protocol [11]", but reference is in Slovak. I suggest including a little explanation about protocol.
    • The text has been modified according to the requirements of the opponent.
  • Line 87: What software?
    • The text has been modified according to the requirements of the opponent.
  • Table 1: Table legend must be include to better understanding. 
    • The text has been modified according to the requirements of the opponent.

Round 2

Reviewer 1 Report

Dear authors,
I consider that all my comments and suggestions had been answered point-by-point. 
I accept the manuscript in this present form to be published.
Congratulations.
Yours faithfully,

Reviewer 2 Report

good job

Reviewer 4 Report

Congratulations